# Determinants of late detection and advanced-stage diagnosis of breast cancer in Nigeria

Olayide Agodirin[1,2]*, Samuel Olatoke[1], Ganiyu Rahman[1,3], Oladapo Kolawole[4], Saliu Oguntola[5], Olalekan Olasehinde[6], Omobolaji Ayandipo[7], Julius Olaogun[8], Aba Katung[9], Amarachukwu Etonyeaku[10], Olufemi Habeeb[1], Ademola Adeyeye[11], John Agboola[2], Halimat Akande[12], Olusola Akanbi[5], Oluwafemi Fatudimu[13], Anthony Ajiboye[2]

1 Department of Surgery, University of Ilorin and University of Ilorin Teaching Hospital, Ilorin, Nigeria, 2 Department of Surgery, General Hospital, Ilorin, Nigeria, 3 Department of Surgery, Cape Coast University and Cape Coast Teaching Hospital, Cape Coast, Ghana, 4 Department of Surgery, LAUTECH Teaching Hospital, Osogbo, Nigeria, 5 Department of Surgery, LAUTECH Teaching Hospital, Ogbomoso, Nigeria, 6 Department of Surgery, Obafemi Awolowo University and Obafemi Awolowo University Teaching Hospital, Ile-Ife, Nigeria, 7 Department of Surgery, University College Hospital, Ibadan, Nigeria, 8 Department of Surgery, Ekiti State University and Ekiti State University Teaching Hospital, Ado-Ekiti, Nigeria, 9 Department of Surgery, Federal Medical Center, Owo, Nigeria, 10 Department of Surgery, Obafemi Awolowo University Ile-Ife and Obafemi Awolowo Teaching Hospital Complex, Ilesha, Nigeria, 11 Department of Surgery, University of Ilorin Teaching Hospital, Ilorin, Nigeria, 12 Department of Radiology, University of Ilorin and University of Ilorin Teaching Hospital, Ilorin, Nigeria, 13 Department of Surgery, Federal Teaching Hospital, Ido-Ekiti and Afe Babalola University, Ado-Ekiti, Nigeria

* cancer1992@yahoo.com

**Data Availability Statement:** All relevant data are within the manuscript and its Supporting information files.

## Abstract

Late detection of Breast cancer(BC) and progressing with advanced-stage diagnosis after early detection contribute differently to the challenges of managing BC in Africa. Understanding the difference may improve cancer education programs and their effectiveness.

### Objective

To describe the risk factors for late detection and advanced-stage diagnosis among patients who detected their BC early.

### Method

Using secondary data, we analyzed the impact of socio-demographic factors, premorbid experience, BC knowledge, and health-seeking pattern on the risk of late detection and advanced-stage diagnosis after early BC detection. Test of statistical significance in SPSS and EasyR was set at 5% using Sign-test, chi-square tests (of independence and goodness of fit), odds ratio, or risk ratio as appropriate.

### Result

Most socio-demographic factors did not affect detection size or risk of disease progression in the 405 records analyzed. High BC knowledge, p-value = 0.001, and practicing breast self-examination (BSE) increased early detection, p-value = 0.04, with a higher probability (OR 1.6 (95% CI 1.1–2.5) of detecting <2cm lesions. Visiting alternative care (RR 1.5(95% CI 1.2–1.9), low BC knowledge (RR 1.3(95% CI 1.1–1.9), and registering concerns for

**Funding:** The African Research Group for Oncology (ARGO) sponsored the data-generating primary research with a pilot grant award to AO. The funders had no role in study design, data collection and analysis, decision to publish, or preparation of the manuscript.

**Competing interests:** The authors have declared no competing interests exist.

hospital care increased the risk of advanced-stage diagnosis after early detection (64% (95% CI 55–72)). Adhering to the monthly BSE schedule reduced the risk of advanced-stage diagnosis by -25% (95% CI -49, -1.1) in the presence of socioeconomic barriers.

## Conclusion

Strategies to increase BC knowledge and BSE may help BC downstaging, especially among women with common barriers to early diagnosis.

## Introduction

Downstaging symptomatic breast cancer (BC) through early detection and diagnosis is a topical issue in sub-Saharan Africa (SSA) because socioeconomic barriers and lack of infrastructure make asymptomatic population-based screening impractical. The patient demographic and presentation patterns also make mammography an ineffective screening tool [1, 2].

Promoting early detection practices and following through with early diagnosis and timely treatment will improve BC outcomes in Africa [1]. Most breast cancers are incidental self-detection in Africa, and up to 80% are diagnosed at stage III or IV, with most patients delaying treatment for more than six months.

There are reports on the risk factors for late presentations in SSA [3–6]. However, direct reports on determinants of late detection among BC patients are rare. The few reports on early detection practices identified in a Black et al. [1] review were in healthcare providers and non-afflicted laywomen. Furthermore, reports on advanced-stage diagnosis failed to distinguish tumors already advanced at the time of detection or recognition from those progressing after early detection. Yet, our previous research showed that late detection and progressing to advanced disease after an early detection contributed differently to advanced-stage diagnosis and challenges of managing BC. In that research, 10% of tumors detected inadvertently were already advanced, while 30–70% of those detected early progressed to advanced-stage before diagnosis [7].

Late detection and progressing after early detection may have different determinants, and understanding the difference might improve cancer education programs' effectiveness in SSA. This study aimed to describe risk factors for late-stage detection and advanced-stage diagnosis among a subpopulation of BC patients who detected their cancers early.

## Materials and method

This research was a secondary data analysis using de-identified data from a study sponsored by the African Research Group for Oncology (ARGO). The original research was a cross-sectional multicentered survey in referral centers in Northcentral and Southwestern Nigeria, including convenience sampling of newly diagnosed BC patients between June 2017 and May 2018 after obtaining ethical approval from all institutions and written consent from participants. The ethical review committee gave additional approval for the current study [UITH ERC PAN/2021/01/0170].

Trained personnel collected the data using specially designed pilot-tested proforma and entered it into a specially designed Microsoft Access database. The primary research instituted mechanisms to minimize recall bias and ensure data reliability. Details of the design and original data collection are available elsewhere [7].

Data of interest in the present research were the demographics and socioeconomic variables, including age, sex, employment status, religion, and marital status. The premorbid experience including the source of BC information, awareness, prior knowledge of BC treatment and outcomes, concern about hospital treatment of BC, and health-seeking pattern: premorbid help-seeking preferences, breast self-examination/ clinical breast examination, number of hospitals, and health care providers (HCP) visited to treat current disease, the initial symptom, tumor size at detection and diagnosis, and reason for referral. The tumor size at detection was a retrospective record, while the size at diagnosis was prospective in the original research.

## Statistical analysis

In this analysis, clinical tumor size (T-size) was the surrogate for disease stage using the T1-3 according to the 7th edition of the American Joint Committee on Cancer (AJCC) staging for BC, where T1 was ≤2cm, T2 was 2.1-5cm, and T3 was >5cm. We defined early detection and early diagnosis as size ≤5cm and small tumor as size ≤2cm. To evaluate each risk factor, we analyzed its impact on the T-size distribution, the odds of detecting small tumors (≤2cm), and the probability of being diagnosed early after early detection.

The effect of age was analyzed by comparing three subgroups; <40, 41–60, and >60. Marital status was analyzed as married vs. unmarried (with unmarried comprising single, widows, separated, or divorced). The level of education was analyzed as secondary/tertiary vs. no education/ primary level. Three subgroups of BC Knowledge were compared; no, low and high knowledge. [We defined none as the lack of BC awareness, low as BC awareness only, and high as BC awareness plus any additional information such as knowing screening modalities, types of breast lumps, BC treatment, or outcome of someone who had BC]. Being employed was compared to being unemployed. Living close to the study center (<30 minutes drive) was compared to living remotely (31–60 minutes and >60 minutes). Consulting an orthodox health care provider first was compared to consulting alternative medicine first. Visiting only one healthcare provider (HCP) was compared to visiting more than one HCP, and visiting a single hospital was compared to visiting multiple hospitals before diagnosis.

The inferential statistic for single proportion used the binomial test, comparison of two or more proportions used the chi-square test of independence/ Fisher's exact or goodness of fit. The risks (probabilities of events) were compared using either the binary logistic regression for the odds ratio or the risk ratio. We presented the results using descriptive statistics, including the 95% confidence intervals (95% CI). We set the test of statistical significance at 5% for all analysis. The analysis was conducted in SPSS v20 and EasyR.

## Results

This research included 405 records of patients aged 24–95 (mean 49.3±16, median 49). The majority were middle age (176,43.4%), married (274, 79%), and educated (175, 43.2%). The younger age groups were more educated (p-value = 0.001). The majority (294, 73%) were aware of BC, 162 (55%) of which had low knowledge, and 132 (45%) had high knowledge. The most common source of BC information was a radio program. The younger patients had more BC information (p-value = 0.02) despite similar information sources across all age groups. Many patients obtaining information from non-medics (person-to-person social contacts and media outlets) had high BC knowledge. Fifty-five percent (95% CI 50–60) of the patients reported practicing breast self-examination (BSE), with only 17% (95% CI 12–23) maintaining the standard monthly schedule (Table 1).

**Table 1. The patient demographics, distribution of information source, the comparison of level of eduction, practice of breast self-examination and information across age groups, comparison of breast cancer knowledge based on information source and education and distribution of BC knowledge across age groups.**

| The patient demographics | | | | | | | |
|---|---|---|---|---|---|---|---|
| | | N(%) | | | | | N(%) |
| Age | 40 & below | 103(25) | | Marital | Married | | 274(68) |
| | 41 to 60 | 176(43) | | | Divorced | | 7(2) |
| | Above 60 | 70(17) | | | Widow | | 43(11) |
| | NS | 56(15) | | | Single | | 21(5) |
| | | | | | NS | | 57(14) |
| Education | None | 44(11) | | Religion | Christian | | 283(70) |
| | Primary | 48(12) | | | Muslim | | 108(27) |
| | Secondary | 57(14) | | | NS | | 11(3) |
| | Tertiary | 118(29) | | Side | left | | 147(37) |
| | NS | 138(34) | | | Right | | 143(36) |
| | | | | | NS | | 112(27) |

| Distribution of information source | | | | | | | |
|---|---|---|---|---|---|---|---|
| Social contact/person-to-person | Church | 13(3.1) | | | | | |
| | Facebook | 9(2.2) | | | | | |
| | Relations/Friends | 41(10) | | | | | |
| | School | 11(3.0) | | | | | |
| Media | Flier | 2(0.5) | | | | | |
| | Newsprint | 3(0.7) | | | | | |
| | Radio | 76(19) | | | | | |
| | Television | 21(5.1) | | | | | |
| Health talk | Hospital/NGO | 49(12) | | | | | |
| | NS | 180(44.4) | | | | | |

| Comparison of the level of education across age groups | | | | | | | |
|---|---|---|---|---|---|---|---|
| | Age distribution (years) | | | p-value | | | |
| | 40 | 41–60 | >60 | | | | |
| Low education | 11 | 43 | 39 | 0.001 | | | |
| High education | 98 | 168 | 47 | | | | |

| Comparison of the practice of Breast Self-Examination across age groups | | | | | | | |
|---|---|---|---|---|---|---|---|
| Practice BSE | 49 | 106 | 28 | 0.01 | | | |
| Not practice | 45 | 73 | 45 | | | | |

| Comparison of the source of information across age groups | | | | | | | |
|---|---|---|---|---|---|---|---|
| Person-to-person | 6 | 35 | 10 | 0.26 | | | |
| media | 20 | 58 | 23 | | | | |
| healthtalk | 13 | 24 | 7 | | | | |

| Comparison of BC knowledge based on the information source | | | | | | | |
|---|---|---|---|---|---|---|---|
| | Person to person | Media | Healthtalk | | | | |
| Low BC knowledge | 30 | 57 | 33 | 0.01 | | | |
| High BC knowledge | 32 | 48 | 11 | | | | |

| Distribution of BC knowledge in low education compared to high education | | | | | | | |
|---|---|---|---|---|---|---|---|
| | Breast cancer knowledge | | | | | | |
| | None | Low | High | | | | |
| Low education | 36 | 36 | 22 | 0.02 | | | |
| High education | 76 | 127 | 109 | | | | |

| Distribution of BC knowledge across age groups | | | | | | | |
|---|---|---|---|---|---|---|---|

(*Continued*)

**Table 1.** (Continued)

| | | | | | | |
|---|---|---|---|---|---|---|
| ≤40 | 42 | 41 | 38 | 0.02 | | |
| 41–60 | 48 | 85 | 78 | | | |
| >60 | 25 | 45 | 20 | | | |

BSE- Breast Self Examination, Divorced = Divorced or separated, NGO- Nongovernmental, NS- Not specified Organization, Person-to-person = social and person-to-person contact.

## Determinants of tumor size distribution at detection

Common social and demographic factors such as age, level of education, marital status, education level, and employment status did not significantly affect tumor size distribution at detection. The premorbid health-seeking behavior, BC information source, and tumor laterality did not significantly affect tumor size distribution at detection (Table 2). More Christians detected earlier (≤5cm) tumors than Muslims (p-value = 0.001). Women with higher knowledge of BC detected earlier (≤5cm) tumor (p-value = 0.001), and women practicing BSE also detected earlier (≤5cm) tumors (p-value = 0.04) (Table 2). The odds of detecting small tumors (≤2cm) were also significantly higher among women practicing BSE (OR 1.6 (95%CI 1.1–2.5)). However, being educated (low education OR 0.9 (95% CI 0.6–1.1), high education OR 1.4 (95% CI 0.8–2.2) did not significantly affect the odds of detecting small tumors ((≤2cm) compared to being uneducated. Also, being a Christian (OR 1.2 (95% CI 0.8–2.0) did not significantly affect the odds of detecting small tumors compared to being a Muslim. More women with high BC knowledge practiced BSE and adhered to the regular monthly schedule. Tumor laterality did not affect BSE's impact on the probability of detecting small tumors.

## Risk factors for tumor progression after early detection

Age, level of education, employment, marital status, and place of residence did not significantly affect the risk of tumor progression after early detection. The pattern of symptomatology, learning about BC from non-medical personnel, and visiting multiple hospitals or multiple health care providers did not significantly affect the risk of tumor progression. First visiting an alternative to orthodox medical care, low BC knowledge, and not practicing BSE were associated with significant risk of progression (Table 3 and Fig 1). Among 79 patients who visited alternative care first, the majority resided close to the referral center (89% (95% CI 80–95). In the same population, 70% of those residing <60 minutes away experienced disease progression whereas a smaller proportion (44% (4 of 9)) of those living remotely experienced disease progression. The difference was not statistically significant (p-value = 0.14). Familiarity with BC patients and knowing poor BC outcomes were not significant deterrents to early-stage diagnosis (Table 3).

## Risk of disease progression after early detection in patients with barriers to early presentation

Subgroup exploratory analysis showed significant risk of disease progression among women expressing any concern (64%(95% CI 55–72). The risk of progression was also significant among those expressing cost concern (67% (95% CI 53–79), whereas the risk was high but not statistically significant among those expressing concern about having a mastectomy (62% (95%CI 42–71) (Table 4 and Fig 2).

**Table 2. The effect of the demographics, premorbid treatment preferences, level of knowledge, and practice of breast-self examination on tumor size distribution at detection.** Also, showing the regularity of Breast Self-Examination based on breast cancer knowledge.

| Effect of age on the distribution of tumor size at detection | | | | | |
|---|---|---|---|---|---|
| | | T1 n(%) | T2 n(%) | T3 n(%) | p-value |
| Age | 40 and below | 52(46) | 51(45) | 11(9) | 0.42 |
| | 41–60 | 80(39) | 102(49) | 24(11) | |
| | Above 60 | 30(34) | 45(51) | 14(15) | |
| Effect of Premorbid treatment preference on the distribution of tumor size at detection | | | | | |
| | Alternative | 8(27) | 14(49) | 7(24) | 0.17 |
| | Self-medicate | 61(46) | 60(44) | 15(10) | |
| | Visit hospital | 71(39) | 92(50) | 20(11) | |
| Effect of religion on the distribution of tumor size at detection | | | | | |
| | Muslim | 39(19) | 57(26) | 111(55) | 0.001 |
| | Christian | 116(40) | 137(46) | 35(14) | |
| Effect of breast cancer knowledge on the distribution of tumor size at detection | | | | | |
| | No knowledge | 42(38) | 60(54) | 10(8) | 0.001 |
| | Low knowledge | 61(36) | 79(49) | 25(15) | |
| | High knowledge | 60(45) | 59(44) | 15(11) | |
| Effect of level of education on the distribution of tumor size at detection | | | | | |
| | Low education | 32(33) | 51(53) | 12(12) | 0.38 |
| | High education | 131(44) | 147(47) | 38(9) | |
| Effect of employment status on the distribution of tumor size at detection | | | | | |
| | Unemployed | 5(31) | 6(38) | 5(31) | 0.14 |
| | Employed | 81(38) | 103(49) | 28(13) | |
| | Unmarried | 25(35) | 34(49) | 12(16) | 0.24 |
| | Married | 114(45) | 135(49) | 28(6) | |
| Effect of practice of Breast Self-Examination on the distribution of tumor size at detection | | | | | |
| Practice BSE | No | 58(36) | 92(57) | 12(7) | 0.04 |
| | Yes | 93(47) | 82(41) | 25(12) | |
| BSE schedule | Daily | 27(54) | 19(38) | 4(8) | 0.03 |
| | Weekly | 4(20) | 9(45) | 7(35) | |
| | Monthly | 14(42) | 16(47) | 4(11) | |
| | Occasionally | 48(47) | 38(42) | 10(11) | |
| Effect of tumor laterality on the distribution of size at detection | | | | | |
| | left | 51(34) | 81(53) | 19(13) | 0.21 |
| | right | 60(41) | 63(43) | 22(16) | |
| Effect of information source on the distribution of tumor size at detection | | | | | |
| | Person to person | 20(32) | 31(49) | 12(19) | 0.42 |
| | Media | 46(42) | 52(47) | 12(11) | |
| | Healthtalk | 21(47) | 20(43) | 5(10) | |
| Comparison of the distribution of tumor size based on the laterality of lesion | | | | | |
| | | T1n (%) | T2n (%) | T3n (%) | p-value |
| Among those performing BSE | left | 25(42) | 27(45) | 8(13) | 0.33 |
| | Right | 35(47) | 24(32) | 14(21) | |
| Among those on regular BSE | Left | 4(29) | 8(57) | 2(14) | 0.32 |
| | Right | 6(55) | 3(27) | 2(18) | |
| Regularity of Breast Self-Examination based on breast cancer knowledge | | | | | |
| | | No know | Low know | High know | |
| BSE regularity | Occasional | 9(10) | 16 (18) | 64(72) | 0.001 |
| | Too frequent | 26(35) | 43(57) | 6(8) | |
| | Standard monthly | 1(3) | 4(12) | 29(85) | |

**Table 3. The effect of the demographics and socioeconomic factors, symptomatology, premorbid preferences, and level of breast cancer information on the risk of disease progression after early detection and the probability of disease progression in the presence of barriers to early presentation.**

| | Progression | No(n) | Yes(n) | p-value | Risk Ratio (95%CI) |
|---|---|---|---|---|---|
| **Age** | | | | | |
| | <40 | 33 | 70 | 0.43 | 1 |
| | 41–60 | 66 | 110 | | 0.9(0.8–1.1) |
| | >60 | 29 | 41 | | 0.9(0.7–1.1) |
| **Level of Education** | | | | | |
| | Educated | 105 | 167 | 0.16 | 1 |
| | Uneducated | 23 | 56 | | 1.2(1.0–1.4) |
| **Religion** | | | | | |
| | Christian | 83 | 162 | 0.44 | 1 |
| | Muslim | 36 | 56 | | 0.9(0.9–1.1) |
| **Distance (Drive to study center in minutes)** | | | | | |
| | 0–30 | 61 | 79 | 0.74 | 1 |
| | 31–60 | 25 | 40 | | 1.1(0.9–1.4) |
| | >60 | 11 | 13 | | 1.0(0.6–1.4) |
| **Marital Status** | | | | | |
| | Married | 85 | 159 | 1.0 | 1 |
| | Unmarried | 20 | 38 | | 1(0.8–1.2) |
| **Employment Status** | | | | | |
| | Employed | 61 | 20 | 0.47 | 1 |
| | Unemployed | 7 | 4 | | 1.4(0.6–3.5) |
| **Tumor Laterality** | | | | | |
| | Right | 42 | 48 | 0.64 | 1 |
| | Left | 81 | 79 | | 1.1(0.9–1.3) |
| **Tumor size** | | | | | |
| | <2cm | 65 | 95 | | 1 |
| | >2cm | 63 | 123 | | 1.1(0.9–1.3) |
| **Symptomatology** | | | | | |
| | lump | 33 | 70 | 0.97 | 1 |
| | No lump | 84 | 171 | | 1.0(0.9–1.2) |
| **Health seeking preference** | | | | | |
| | Visit hospital | 68 | 89 | 0.007 | 1 |
| | Self-medicate | 36 | 83 | | 1.2(1–1.5) |
| | Alternative care | 3 | 18 | | 1.5(1.2–1.9) |
| **Number of Hospital or HCP visited** | | | | | |
| | 1hospital | 48 | 70 | 0.33 | 1 |
| | >1hospital | 16 | 35 | | 1.1(0.9–1.5 |
| HCP | 1HCP | 44 | 61 | 0.37 | 1 |
| | >1HCP | 66 | 118 | | 1.1 (0.9–1.3) |
| **Breast Self Examination** | | | | | |
| BSE | perform | 77 | 83 | 0.0003 | 1 |
| | Not Perform | 40 | 107 | | 1.4(1.2–1.7) |
| Regularity | Monthly | 16 | 13 | 0.78 | 1 |
| | Weekly | 7 | 6 | | 1.0(0.5–2) |
| | Daily | 19 | 21 | | 1.2(0.7–2) |
| | Occasionally | 35 | 43 | | 1.2(0.8–2) |
| **Source of BC information** | | | | | |

*(Continued)*

**Table 3.** (Continued)

| | | | | | |
|---|---|---|---|---|---|
| e | Health talk | 21 | 20 | 0.31 | 1 |
| | Others | 99 | 141 | | 1.2(0.9–1.7) |
| **Level of Breast Cancer Knowledge** | | | | | |
| | High knowledge | 56 | 60 | 0.0002 | 1 |
| | No knowledge | 21 | 79 | | 1.5(1.2–1.9) |
| | Low knowledge | 51 | 84 | | 1.3(1.1–1.5) |
| **Previous Interaction With BC Patients** | | | | | |
| Interaction | No patient known | 5 | 2 | 0.23 | 1 |
| | Knows BC patient | 13 | 19 | | 2.0(0.6–7.0) |
| Outcome known | Alive | 4 | 5 | 1.0 | 1 |
| | Died | 9 | 12 | | 1.0(0.5–2.0) |
| **Prompt for Visiting Specialist** | | | | | |
| Reason | referred | 42 | 73 | 0.57 | 1 |
| | self | 11 | 27 | | 1.1(0.9–1.4) |
| | advice | 14 | 33 | | 1.0(0.8–1.3 |
| **The probability of disease progression in the presence of barriers to early presentation** | | | | | |
| | Subgroups with Barrier (Concerns) | No(n) | Yes(n) | | Risk of Progression in subgroup% (95% CI) |
| | Concern for mastectomy | 21 | 34 | 0.11 | 62 (48–75) |
| | Mastectomy concern in forty years and below | 7 | 15 | 0.13 | 68 (45–86) |
| | Mastectomy concern above 40 years | 14 | 19 | 0.48 | 58 (39–75) |
| | Cost concern | 21 | 41 | 0.015 | 66 (53–78) |
| | Other concerns | 11 | 15 | 0.56 | 58 (37–77) |

HCP- Health Care Provider.

Other concerns: The attitude of personnel, chemotherapy/fertility, conflicting statements, delay/bureaucracy/stress, death.

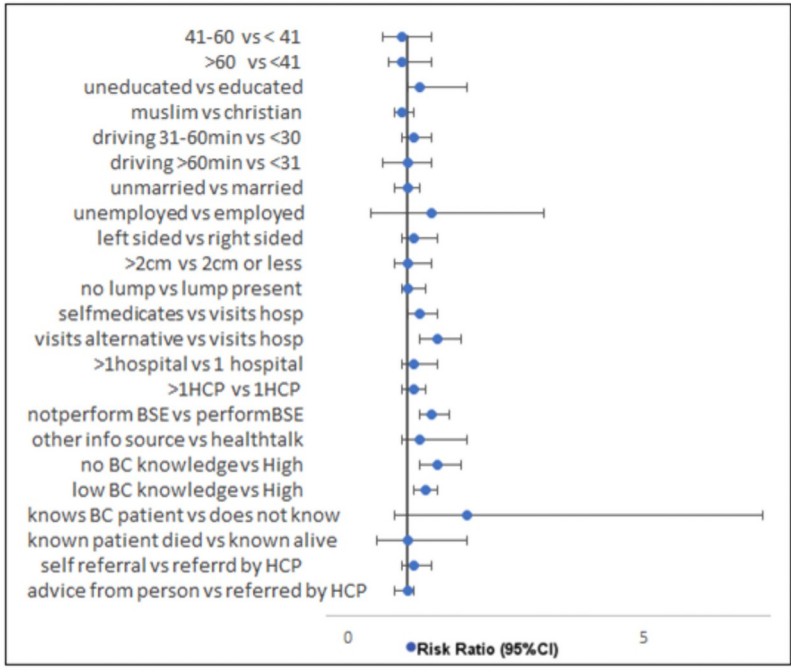

**Fig 1. Forest plot of risk ratio for disease progression.** Showing the Risk Ratio for disease progression based on demographic, socioeconomic factors, symptomatology premorbid preferences, and knowledge. BC- Breast Cancer, BSE-Breast Self-Examination, HCP-Healthcare Provider, vs = 'compared to'.

**Table 4. Showing interaction between practice of BSE and incidence of progression.**

| Risk of progression among patients with barrier | | | | |
| --- | --- | --- | --- | --- |
| Concern (N) | No progression | Progression | Risk of Progression (95%CI) | p-value |
| Any (141) | 51 | 90 | 64(55–72) | 0.001 |
| Mastectomy (49) | 21 | 28 | 57(42–71) | 0.39 |
| Cost (57) | 19 | 38 | 67(53–79) | 0.016 |
| Others (35) | 11 | 24 | 69(51–83) | 0.041 |
| Among those practicing any BSE | | | | |
| Any concern (74) | 33 | 41 | 54(43–67) | 0.42 |
| Mastectomy (27) | 14 | 13 | 49(29–68) | 1.0 |
| Cost (28) | 13 | 15 | 53(34–75) | 0.85 |
| Others (19) | 6 | 13 | 68(43–87) | 0.17 |
| Among those practicing monthly BSE | | | | |
| Any concern (18) | 11 | 7 | 39(17–64) | 0.48 |
| Mastectomy (7) | 6 | 1 | 14(0.4–58) | 0.13 |
| Cost (60 | 3 | 3 | 50(12–88) | 1.0 |
| Others (5) | 2 | 3 | 60(15–95) | 1.0 |
| Comparison between those practicing BSE and those not practicing | | | | |
| | No BSE N(progession) | Practice BSE N(progression) | Risk Ratio for progression | |
| Any | 67(49) | 74(41) | 1.32(1.0–1.7) | 0.04 |
| Mastectomy | 22(15) | 27(13) | 1.4(0.9–2.3) | 0.25 |
| Cost | 29(23) | 28(15) | 1.5(1.0–2.2) | 0.05 |
| Others | 16(11) | 19(13) | 1.0(0.6–1.6) | 1.0 |

BSE- Breast Self Examination.

Progression = number progressing from early to an advanced stage before diagnosis.

Risk of progression = number progressing divided by the total number (N).

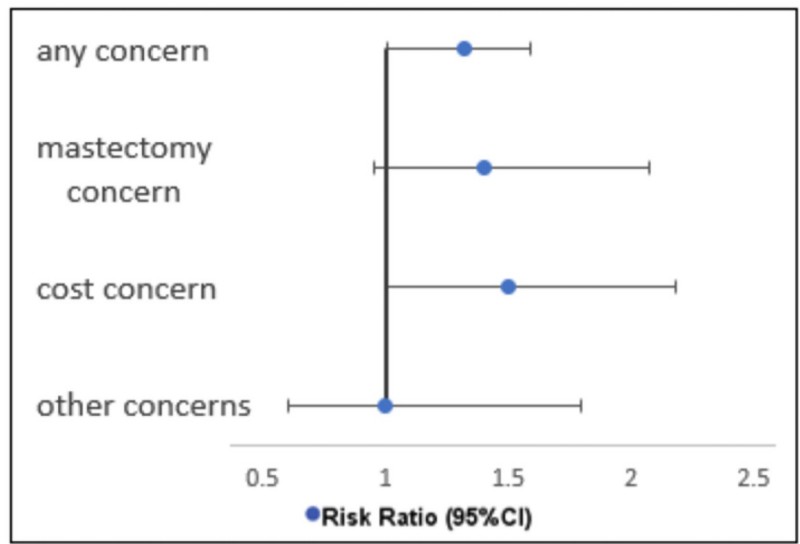

**Fig 2. Forest plot of risk ratio of disease progression among those performing BSE.** Showing Risk Ratio of disease progression among patients not performing Breast Self-Examination compared to those performing Breast Self-Examination.

In patients expressing any concern for hospital care, the risk of disease progression was lower among those practicing BSE (-8.5% (95% CI -22, 5.4) compared to not practicing BSE and more so among those adhering to a monthly routine, with a significantly reduced risk of -25% (95% CI -49, -1.1).

## Discussion

Breast cancer causes significant morbidity and mortality in SSA, and much-needed information on late detection and factors associated with advanced-stage diagnosis to improve BC outcomes in SSA is scarce. We researched the risk factors for late detection and advanced stage diagnosis of the disease after early detection. We found that high BC knowledge and practicing BSE were associated with detecting smaller tumors and lowered the risk of progressing to advanced stages before diagnosis. Visiting alternatives to orthodox care and concerns about healthcare costs were associated with disease progression.

The initial breast lump detection mode has prognostic implications; lesions detected by mammography are often smaller with a better prognosis than those seen by BSE [8]. Our finding here suggests that women practicing BSE detected small lumps. BSE's role is controversial as studies in developed countries [9, 10] showed that it increased the frequency of breast complaints, prompted more visits to physicians, and led to more biopsies without significant benefits in tumor size at detection or survival [11]. Nonetheless, the evidence supporting BSE/ clinical breast examination (CBE) is mounting, especially in centers where BC is detected inadvertently and diagnosed at the late stages [12–15].

Significant gaps exist in our knowledge of how best to improve early presentation and acceptance of BC treatment in SSA. Since population-based screening is not feasible [16] due to economic, infrastructural, and personnel deficiencies, Africa must explore innovative low-cost, and sustainable means of downstaging the disease. A program [12] in Sudan's rural communities used volunteers in door-to-door breast examination to increase early BC detection. Another research to integrate breast health services into clinical practice in Ghana [17] proposed a model grounded in human interaction and based on the experience of BC patient and their relations whereby trained personnel offered breast cancer information, the teaching of BSE, and CBE to the relations and micro-communities of breast cancer patients [17].

Given that most breast cancer in Africa is self-detected, understanding the BSE barriers and determinants is essential. Factors reported to affect BSE performance are years of college education, knowledge of BSE and its method, perception of the benefit of associating BSE with a likelihood of detecting smaller lesions, longer breastfeeding duration, the pressure of responsibilities, and forgetfulness [18–20].

Strategies to downsize Breast cancer rely on widespread patient-level education, personnel training, and an organized healthcare system [13] to retain patients and complete treatment. Unfortunately, ensuring timely diagnosis and adhering to medical care after disease detection is still a challenge in Nigeria and Africa. Encouragingly, we found that practicing BSE increased the chances of early diagnosis despite known barriers, even among those registering concern for mastectomy and cost of hospital treatment. This suggests that practicing BSE might not only influence early detection; it might also be a predictor of the willingness or motivation to follow through with timely diagnosis and treatment, with the strongest association among women practicing BSE in the standard monthly schedule.

The influence of the common socio-demographic factors on delay varies remarkably within and across regions of Africa. Such factors as age, level of education, marital status, residing remotely, and employment status did not influence detection size and risk of disease

progression in the cohort studied here. Nevertheless, these factors merit further research as they often contribute to delay and treatment challenges in SSA.

The influence of symptomatology on delay is fairly consistent in the literature; the absence of breast lump, pain, and ulceration are linked to delays. Recently, the implication of laterality is becoming more apparent. A report in India [21] found right-sided tumors were diagnosed at later stages compared to left-sided tumors, suggesting the impact of handedness. It is reassuring in this study that tumor laterality did not appear to diminish the effect of BSE on detecting small tumors.

Another possibly advantageous exploratory finding pertinent to the challenges of managing BC in poor-resource centers is that receiving information from non-medics might not negatively affect its benefit. Getting comprehensible breast health messages to as many women as possible might be more important than the source. Similarly, A recent report from Uganda found that irrespective of the source, women who received breast health education previously participated more in BSE and CBE [22]. However, there was segregation depending on the place of health care service, with women receiving care in public services preferring messages from healthcare providers. In contrast, those paying out of pocket preferred messages from friends and family.

Africa needs more context-specific interventional research similar to the effort in Sudan [12] using locally trained personnel for CBE and in Ghana [17] intervening on BC patients' micro-communities using CBE and BSE. Such studies should assess the feasibility, cost-effectiveness, and benefit of the BC down-staging strategies. Simple assumptions or over-generalization of research findings should be avoided in rolling out interventional programs or policies because of unexpected and counterintuitive findings. For instance, family history of BC and history of benign lesions were associated with increased risk of endstage disease [21], negating the expected positive effect of prior knowledge in an Iranian study. Our findings showed significant use of alternatives and a higher proportion of disease progression among those living close to referral centers, thus negating the expected effect of distance.

Being a secondary analysis limits our findings. Also, we did not directly determine that the tumors were detected during BSE, and we did not assess the knowledge of BSE and the method. Furthermore, we could not evaluate the association between detection, time to treatment, and treatment outcome. Notwithstanding, the present report is one of few studies on factors associated with advanced-stage diagnosis in Africa, providing insight into some previously unreported and under-researched associations that might aid down-staging breast cancer in Africa.

## Conclusion

Most of the socioeconomic and demographic risk factors commonly influencing late presentation and diagnosis of BC in Africa did not affect early detection or risk of progression among patients who detected their disease early. High knowledge of BC and practicing BSE were consistently associated with early detection and early diagnosis. Additionally, the simple habit of checking the breast might increase early BC detection while adhering to the standard routines of BSE might be associated with detecting even smaller tumors and following through with early diagnosis in the face of common barriers.

## Supporting information

**S1 Data. Minimum data file.**
(XLSX)

## Author Contributions

**Conceptualization:** Olayide Agodirin, Samuel Olatoke, Ganiyu Rahman.

**Data curation:** Olayide Agodirin.

**Formal analysis:** Olayide Agodirin.

**Methodology:** Olayide Agodirin, Samuel Olatoke, Ganiyu Rahman.

**Project administration:** Olayide Agodirin.

**Visualization:** Samuel Olatoke.

**Writing – original draft:** Olayide Agodirin, Samuel Olatoke, Ganiyu Rahman, Oladapo Kolawole, Saliu Oguntola, Olalekan Olasehinde, Omobolaji Ayandipo, Julius Olaogun, Aba Katung, Amarachukwu Etonyeaku, Olufemi Habeeb, Ademola Adeyeye, John Agboola, Halimat Akande, Olusola Akanbi, Oluwafemi Fatudimu, Anthony Ajiboye.

**Writing – review & editing:** Olayide Agodirin, Samuel Olatoke, Ganiyu Rahman, Oladapo Kolawole, Saliu Oguntola, Olalekan Olasehinde, Omobolaji Ayandipo, Julius Olaogun, Aba Katung, Amarachukwu Etonyeaku, Olufemi Habeeb, Ademola Adeyeye, John Agboola, Halimat Akande, Olusola Akanbi, Oluwafemi Fatudimu, Anthony Ajiboye.

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
