## [Decision Letter · Decision Letter 0]

29 Jun 2021

PONE-D-21-10135

Determinants of late detection and advanced-stage diagnosis of breast cancer in Nigeria

PLOS ONE

Dear Dr. Olayide,

Thank you for submitting your manuscript to PLOS ONE. After careful consideration, we feel that it has merit but does not fully meet PLOS ONE’s publication criteria as it currently stands. Therefore, we invite you to submit a revised version of the manuscript that addresses the points raised during the review process.

We look forward to receiving your revised manuscript.

Kind regards,

Shah Md Atiqul Haq

Academic Editor

PLOS ONE

Additional Editor Comments:

Dear authors,

I would ask you to address the issues raised by the reviewers.

Best wishes,

Journal Requirements:

3. In ethics statement in the manuscript and in the online submission form, please provide additional information about the database used in your retrospective study. Specifically, please ensure that you have discussed whether all data were fully anonymized before you accessed them and/or whether the IRB or ethics committee waived the requirement for informed consent. If patients provided informed written consent to have their data used in research, please include this information.

Reviewers' comments:

Reviewer's Responses to Questions

**Comments to the Author**

1. Is the manuscript technically sound, and do the data support the conclusions?

Reviewer #1: Partly

Reviewer #2: Yes

2. Has the statistical analysis been performed appropriately and rigorously? 

Reviewer #1: No

Reviewer #2: Yes

3. Have the authors made all data underlying the findings in their manuscript fully available?

Reviewer #1: No

Reviewer #2: Yes

4. Is the manuscript presented in an intelligible fashion and written in standard English?

Reviewer #1: No

Reviewer #2: Yes

5. Review Comments to the Author

Reviewer #1: Overall: thank you to the authors for presenting this piece of work. I would however, recommend that more work and care is taken to proof read the manuscript and the way the results are presented.

1. Spelling and grammar for example affiliations should be corrected to “Department” not “department”.

- “… socioeconomic barriers and lack of infrastructure make asymptomatic population-based screening impractical. And, patient demographic and presentation patterns make mammography an ineffective screening tool” – use of ‘and’ at the beginning of the sentence

- Revise where the full stops are placed within the sentence e.g. page 13 full stops are placed before (Table 3) and after as well

2. Results need re-working. As the tables stand now it is hard to understand what is being presented:

- Why are odds ratios presented only for some variables in table 2 and not for all

- Table 1 requires a has marital status on the side without any headings

- The use of (a), (b) etc. in the tables is better replaced by actual sub-headings in the tables so that the reader is not required to go back and forth

- Some p-values are written as p-value = … and others just has the numbers – please be consistent

- What are the p-values for table 3? Symptomatology does not have the reference value of 1 in the table

- What is the risk of progression i.e. how did you arrive at that percentage?

3. Not sure what the legend is supposed to show on page 22

Reviewer #2: 1. percentage of variables are missing in the tables. Authors should provide the percentages in the tables.

2. Page 7: Authors wrote "Fischer’s exact" . It should correct as " Fisher's exact test".

3. Table 3 and 4 are helpful to understand the progression.However, additional to table 3 and table 4, forest plot for OR and RR could be helpful to see risk and to visualize.

6. PLOS authors have the option to publish the peer review history of their article (what does this mean?). If published, this will include your full peer review and any attached files.

Reviewer #1: No

Reviewer #2: No

---

## [Author Response · Author response to Decision Letter 0]

7 Aug 2021

We thank the reviews and editor for taking the time and effort to review and suggest corrections in our manuscript. The suggestions have greatly improved the manuscript.

Kind regards 

Reviewer 1 comment Review made in Manuscript Location in manuscript

1 Spelling and grammar corrections Spelling, grammar, and punctuations reviewed throughout the article Throughout the article

 Use of “And” to begin sentence not appropriate Sentence reviewed manuscript line 64)

2 Results need re-working to make tables more understandable Result section and all tables reviewed to simplify Results section

 Why are odds ratio presented for only a few variables in table 2 Odds ratio presented only for variables with statistically significant analysis. However, the odds ratio has been moved to prose section of the result to maintain uniformity Manuscript line 156-169

 Use of letters better replaced with actual headings in tables Letters replaced with actual headings describing each subsection in tables In tables 

 Presenting of the p-values not uniform p-values presented uniformly through out the article Results section

 What is the risk of progression, how did you arrive at that percentage Progression is the number progressing from early to an advanced stage before diagnosis

Risk of progression is number progressing divided by the total number (N)

 Description inserted in footnote of table 4

 What the legend in page 22 mean They are just legends of the tables, they have been removed 

Reviewer 2 

1 Percentages missing in tables Percentages included where appropriate in all tables 

2 Fischer’s exact should be corrected as Fisher’s exact Corrected Manuscript line 129

3 Present forest plots for the OR and RR values in tables 3 and 4 Forest plots presented as figures 1 and 2 

Journal requirements 

 Review reference list Reference list reviewed and corrected; retracted manuscript not used 

 Ensure manuscript conforms with journal style Headings reviewed to conform to journal style 

 Make data available Minimum data available in excel spread sheet 

 Provide additional information about database used The database was a local simple Microsoft database produced specially for the purpose of the original research Manuscript line 94

 Give information on whether data was deidentified and if IRB gave waiver De identified data was used and IRB approval was obtained Manuscript lines 87 and 92

---

## [Decision Letter · Decision Letter 1]

18 Aug 2021

Determinants of late detection and advanced-stage diagnosis of breast cancer in Nigeria

PONE-D-21-10135R1

Dear Dr. Olayide Agodirin,

We’re pleased to inform you that your manuscript has been judged scientifically suitable for publication and will be formally accepted for publication once it meets all outstanding technical requirements.

Kind regards,

Shah Md Atiqul Haq

Academic Editor

PLOS ONE

Additional Editor Comments (optional):

Dear authors,

Congrtaulations!

The paper is accepted.

Best wishes,

Reviewers' comments:

Reviewer's Responses to Questions

**Comments to the Author**

1. If the authors have adequately addressed your comments raised in a previous round of review and you feel that this manuscript is now acceptable for publication, you may indicate that here to bypass the “Comments to the Author” section, enter your conflict of interest statement in the “Confidential to Editor” section, and submit your "Accept" recommendation.

Reviewer #2: All comments have been addressed

2. Is the manuscript technically sound, and do the data support the conclusions?

Reviewer #2: Yes

3. Has the statistical analysis been performed appropriately and rigorously? 

Reviewer #2: Yes

4. Have the authors made all data underlying the findings in their manuscript fully available?

Reviewer #2: Yes

5. Is the manuscript presented in an intelligible fashion and written in standard English?

Reviewer #2: Yes

6. Review Comments to the Author

Reviewer #2: (No Response)

7. PLOS authors have the option to publish the peer review history of their article (what does this mean?). If published, this will include your full peer review and any attached files.

Reviewer #2: No

---

## [Editor Report · Acceptance letter]

1 Sep 2021

PONE-D-21-10135R1 

Determinants of late detection and advanced-stage diagnosis of breast cancer in Nigeria 

Dear Dr. Agodirin:

I'm pleased to inform you that your manuscript has been deemed suitable for publication in PLOS ONE. Congratulations! Your manuscript is now with our production department. 

Kind regards, 

on behalf of

Dr. Shah Md Atiqul Haq 

Academic Editor

PLOS ONE